# Viral gene drive in herpesviruses

Marius Walter [1✉] & Eric Verdin [1✉]

Gene drives are genetic modifications designed to propagate in a population with high efficiency. Current gene drive strategies rely on sexual reproduction and are thought to be restricted to sexual organisms. Here, we report on a gene drive system that allows the spread of an engineered trait in populations of DNA viruses and, in particular, herpesviruses. We describe the successful transmission of a gene drive sequence between distinct strains of human cytomegalovirus (human herpesvirus 5) and show that gene drive viruses can efficiently target and replace wildtype populations in cell culture experiments. Moreover, by targeting sequences necessary for viral replication, our results indicate that a viral gene drive can be used as a strategy to suppress a viral infection. Taken together, this work offers a proof of principle for the design of a gene drive in viruses.

[1] Buck Institute for Research on Aging, Novato, CA 94945, USA. ✉email: mwalter@buckinstitute.org; everdin@buckinstitute.org

Herpesviruses are universal pathogens that are implicated directly or indirectly in numerous human diseases[1]. In particular, human cytomegalovirus (hCMV) is an important threat to immunocompromised patients, such as HIV-infected individuals, receivers of organ transplants and unborn infants. Moreover, life-long hCMV infection in the elderly has been associated with chronic inflammation and immunosenescence, and hCMV could represent an important risk factor in many age-related diseases[2]. While treatment options exist, drug resistance and adverse secondary effects render the development of new therapeutic solutions necessary.

Gene drive refers to the transmission of specific genetic sequences from one generation to the next with a high probability and can propagate a trait to an entire population[3–9]. Most recent engineered gene drive strategies rely on CRISPR-Cas9 editing, where a *Cas9* transgene is inserted in place of a natural sequence, alongside a guide RNA (gRNA) targeting the very same location. During sexual reproduction, repair of an unmodified allele by homologous recombination after cleavage by Cas9 leads to duplication of the synthetic sequence. This ensures that in every generation offspring harbor two copies of the modification and enables the efficient spread of the transgene in the population. The strategy relies on the simultaneous presence of a wildtype and a gene drive allele in the same cell nucleus. For these reasons, it has generally been assumed that a gene drive could only be engineered in sexually reproducing organisms, which excludes bacteria and viruses[7,10].

However, during viral replication multiple copies of the viral genome accumulate in infected cells, effectively rendering the viral genome highly multiploid. Herpesviruses are nuclear-replicating DNA viruses with large dsDNA genomes (100–200 kb) that encode 100–200 genes[11]. They frequently undergo homologous recombination during their replication cycle and can be efficiently edited by CRISPR-Cas9[12–15]. These properties enabled the design of a gene drive strategy that does not involve sexual reproduction, but relies on coinfection of a given cell by a wildtype and an engineered virus (Fig. 1). Upon coinfection, the wildtype genome is cut and repaired by homologous recombination, producing new gene drive viruses that progressively replace the wildtype population.

Here, we present a proof of concept for such a phenomenon, using hCMV as a model. We demonstrate that gene drive viruses can replace their wildtype counterpart and spread in the viral population in cell culture experiments. Moreover, by inserting the gene drive cassette in place of a critical viral gene, we show that the infectivity of the modified virus can be drastically reduced.

## Results

### Generation of gene drive viruses.
We first aimed to build a gene drive system that would not affect viral infectivity, so that it could potentially spread easily into a wildtype population. We chose to insert a gene drive cassette into *UL23*, a viral gene that is dispensable for hCMV replication in human fibroblasts[16]. We cloned a donor plasmid containing homology arms, *Cas9* (from *Streptococcus pyogenes*), an *mCherry* fluorescent reporter and a gRNA targeting *UL23* 5′UTR (Fig. 2a). In this system, *Cas9* transcription is driven by the *UL23* endogenous viral promoter (Fig. 2b). Human foreskin fibroblasts were transfected with the gene drive plasmid and infected with TB40/E-bac4, a laboratory hCMV strain[17]. mCherry-expressing viruses created by homologous recombination were isolated and purified until a pure population of gene drive virus (GD-mCherry) could be obtained (Supplementary Fig. 1a, b). We similarly incorporated an *mCherry* reporter into a neutral region of TB40/E-bac4 (referred hereafter as simply TB40/E, Fig. 2c, Supplementary Fig. 1c). GD-mCherry replicated with a slightly slower dynamic that TB40/E but ultimately reached similar titers (Fig. 2d).

### Recombination of the gene drive into the wildtype genome.
To determine if the gene drive virus could recombine with an unmodified virus, fibroblasts were co-infected with equal quantities of GD-mCherry and an eGFP-expressing virus from a different viral strain (Towne-eGFP)[18], at a multiplicity of infection (MOI) of 1 for each virus. The supernatant of co-infected cells was then used to infect fresh fibroblasts at a very low MOI (<0.001) (Supplementary Fig. 2a, b). In this second generation, we detected cells and viral plaques expressing eGFP alone, mCherry alone, or mCherry and eGFP together (Fig. 3a, Supplementary Fig. 2c). The presence of mCherry-eGFP-expressing viral plaques suggested that both *mCherry* and *eGFP* were present in the same viral genome.

To confirm this hypothesis, we first recovered episomal viral DNA from infected cells using a modified HIRT DNA extraction method[19]. Both GD-mCherry and Towne-eGFP viral genomes

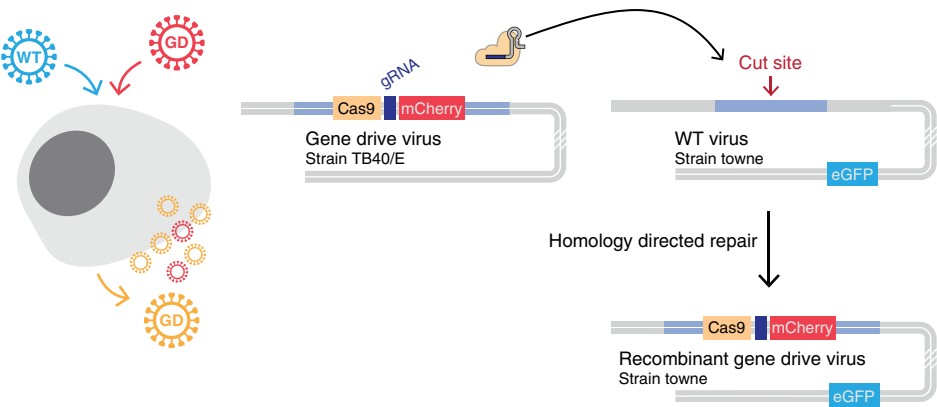

**Fig. 1 Gene drive in herpesviruses.** CRISPR-based gene drive sequences are, at a minimum, composed of *Cas9* and a gRNA targeting the complementary wildtype (WT) locus and can harbor an additional 'cargo' that will be carried over with the rest of the sequence. When present in the same cell nucleus, Cas9 targets and cleaves the WT sequence. Homology-directed repair of the damaged WT locus using the gene drive sequence as a repair template ensures the conversion of the WT locus into a new gene drive sequence. In herpesviruses, gene drive involves the coinfection of a given cell by a WT and a modified virus. Cleavage and repair of the WT genome convert the virus into a new gene drive virus, spreading the modification into the viral population. In this example, hCMV WT virus (Towne strain) expresses eGFP florescent protein, and the gene drive virus (TB40/E strain) carries mCherry. Recombinant viruses then express both eGFP and mCherry.

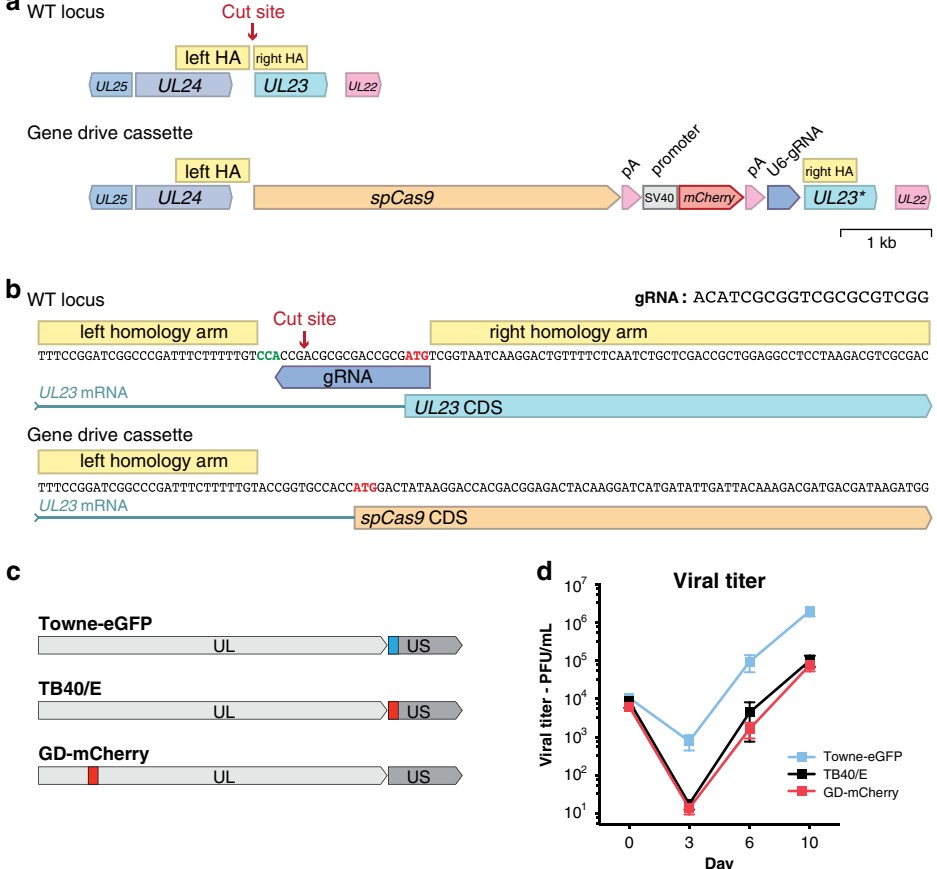

**Fig. 2 Design of a gene drive system targeting hCMV *UL23*. a** Modified and unmodified *UL23* locus. Here, the gene drive cassette is composed of *spCas9* followed by the SV40 polyA signal, an SV40 promoter driving an *mCherry* reporter, the beta-globin polyA signal and a U6-driven gRNA. **b** *UL23* CRISPR cutting site and sequence of WT and modified viruses. *spCas9* transcription is driven by *UL23* viral promoter. CDS: coding sequence. **c** Localizations of *mCherry* and *eGFP* cassette on hCMV genomes. UL/US: unique long/short genome segments. **d** Replication of Towne-eGFP, TB40/E, and GD-mCherry viruses. Viral titers were measured by plaque assay over time. Titers are expressed in plaque forming unit (PFU) per mL of supernatant. Error bars represent standard error of the mean (SEM) between biological replicates. $n = 5$ (TB40/E), or $n = 8$ (Towne-eGFP and GD-mCherry). Source data are provided as a Source Data file.

carry BAC vector sequences that can be used to transform circular viral DNA into competent *E. coli*[17,18]. After transformation of episomal DNA from two independent coinfection experiments, 48 clones were isolated, each representing a distinct viral genome. Importantly, PCR revealed that 19 clones (40%) carried both *eGFP* and *mCherry*, whereas 21 clones (44%) were positive for *mCherry* only and only two clones (4%) for *eGFP* only (Fig. 3b). The presence of multiple *eGFP-mCherry*-positive clones and the relative absence of *eGFP*-only suggested that most of Towne-eGFP viral genomes had incorporated the gene drive cassette containing *mCherry*.

As a control experiment, we coinfected cells with GD-mCherry and Towne-eGFP viruses carrying a mutated target site, thereby preventing Cas9-specific cleavage (see below, Fig. 4b and Supplementary Fig. 7). In this case, we only observed very few mCherry-eGFP-expressing viral plaques from the second generation of viruses. PCR analysis of 30 HIRT-extracted viral genomes did not detect any recombination events, with the majority of clones (83%) expressing *eGFP* only (Fig. 3c). In similar experiments using a *Cas9*-deleted version of the gene drive virus (GD-ΔCas9), we also observed very limited recombination in second generation viruses (see below). These results indicated that the highly efficient incorporation of the mCherry cassette into unmodified Towne-eGFP genomes necessitated both Cas9 and a specific gRNA.

Towne and TB40/E viral strains are differentiated by numerous single nucleotide polymorphisms (SNPs), allowing us to pinpoint homologous recombination breakpoints. Sanger sequencing and analysis of SNPs around the gene drive cassette in recombinant eGFP-mCherry clones showed that in 4 clones out of 17, homologous recombination had occurred immediately next to the CRISPR cut site (Fig. 3d, Supplementary Fig. 3). This also highlighted that in most cases, homologous recombination breakpoint was located more than 1–2 kb away from the insertion site.

We therefore sought to use long-read sequencing to further investigate recombination between gene drive viruses (TB40/E strain) and unmodified viruses (Towne strain). Fibroblasts were co-infected (MOI = 0.1 for both viruses) for 2 weeks, and virions collected from the culture supernatant. Linear viral DNA was extracted from purified virions and subjected to long-read sequencing using Oxford Nanopore sequencing. In two biological replicates, 41,779 and 9926 reads with a length >10 kb were recovered, of which 98.9% and 96.6%, respectively, could be mapped onto the hCMV genome (sequencing summary in Supplementary Fig. 4). For each position along the hCMV genome, we plotted the proportion of SNPs originating from one strain or the other (Fig. 3e, Supplementary Fig. 5a, b). As expected, about 80% of SNPs immediately around the target site originated from the TB40/E donor strain, and this proportion

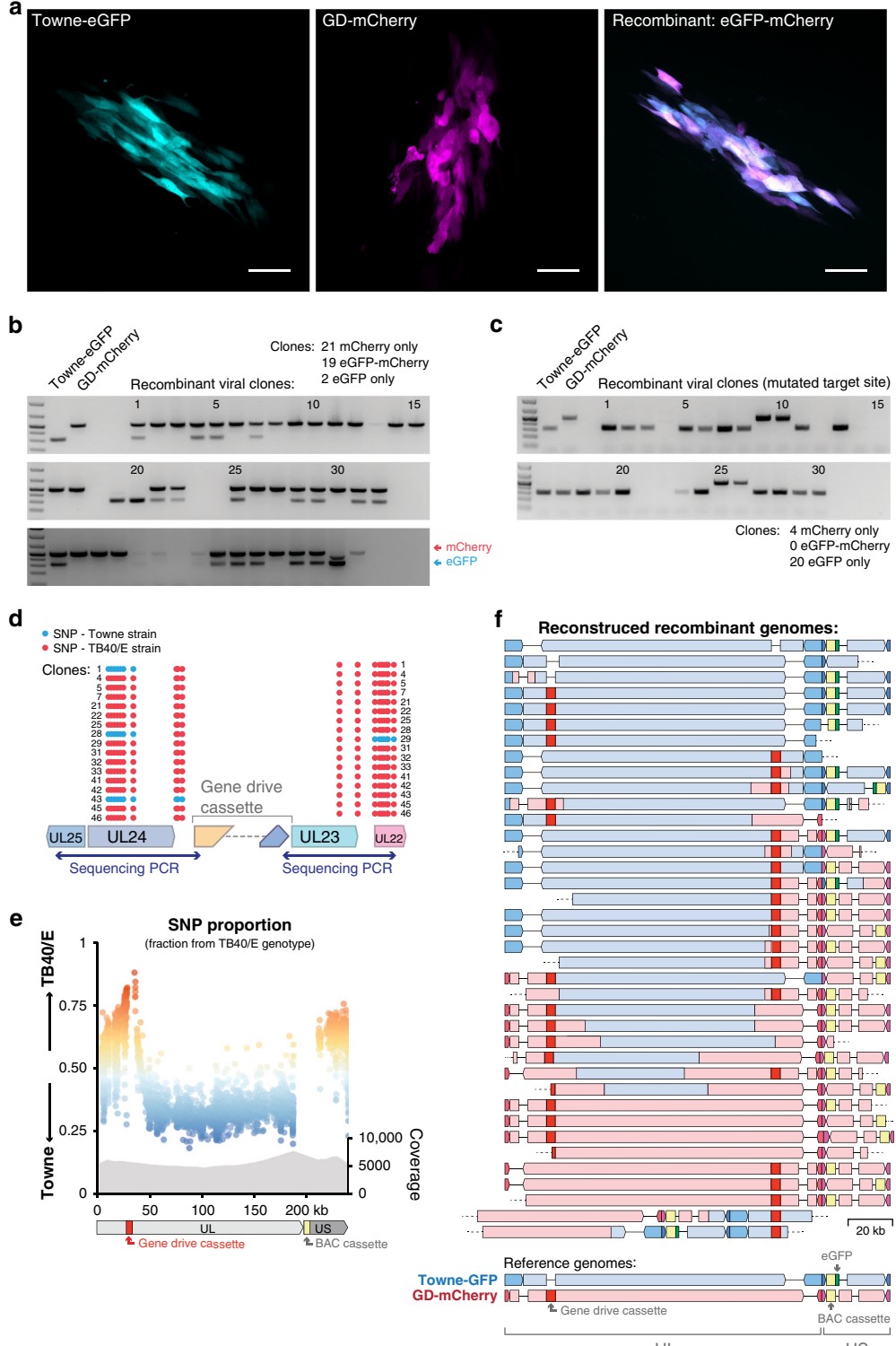

decreased sharply further away from the cutting site. This finding indicated that the gene drive cassette was selectively integrated into the Towne-eGFP genome, most likely because of CRISPR-mediated homologous recombination.

Interestingly, recombination did not occur symmetrically on both sides of the insertion site (Fig. 3e). The hCMV genome comprises two domains, each made of unique segments (named UL for the longer segment, US for the shorter one) flanked by inverted repeats[11,20]. The two domains can be inverted relative to one another, forming four different linear configurations that can

be isolated in equal amounts from viral particles (Supplementary Fig. 5c). The gene drive cassette is located near one extremity of the UL fragment, and the *eGFP* and the BAC cassette are in the US segment. When mapping sequencing reads into the four linear genome configurations, we observed a similar pattern: SNPs around the gene drive cassette and in most of the US segment originated predominantly from the donor TB40/E strain, but SNPs from the remaining of UL segment came predominantly from the Towne strain (Fig. 3e, Supplementary Fig. 5a, b). The enrichment for TB40/E SNPs in the US segment was unexpected.

**Fig. 3 Recombination of the gene drive cassette into the wildtype genome.** Fibroblasts were coinfected with Towne-eGFP (WT) and GD-mCherry, and supernatant was used to infect fresh cells. **a** Representative examples of fluorescent viral plaques spreading on fibroblasts, expressing either eGFP only (left), mCherry only (middle), or both (right). Scale bar: 100 µm. **b** PCR for mCherry (upper band) and eGFP (lower band) on 48 recombinant viral genomes from two independent experiments. **c** Same as (**b**), after coinfection with GD-mCherry and a mutated Towne-eGFP (two biological replicates, 30 genomes). **d** PCR of homology arms and Sanger sequencing of 17 eGFP-mCherry expressing viral clones. Blue dots: single nucleotide polymorphisms (SNPs) from Towne strain; Red dots: TB40/E strain. **e** Fraction of SNPs of Towne or TB40/E origin alongside the hCMV genome. Each dot represents an individual SNP. Data combine two biological replicates after Oxford Nanopore sequencing. Coverage gives the number of reads, allowing multiple mapping. **f** Reconstruction of recombination history of individual hCMV genomes from long (>200 kb) Nanopore reads. One genome corresponds to one sequencing read and colors indicate the strain of origin. Gaps represent regions deleted compared to the reference genome, dashed lines indicate uncovered region. UL/US: unique long/short genome segments. GeneRuler 1 kb DNA ladder (ThermoFisher) was used in agarose gels as a size marker. The ladder is shown in the gel images, with the stronger band corresponding to 500 bp. Source data are provided as a Source Data file.

The genome of herpesviruses is linear, but circularizes after infection in the host cell nucleus, and generates long concatemeric DNA molecules during viral DNA replication[11]. Recombination is an integral part of viral replication and is thought to occur between these circular or concatemeric replication intermediates[21,22]. Circular and concatemeric genomes exist in two configurations that both put the gene drive cassette in relative proximity to the US segment (Supplementary Fig. 5c). This, along with differences in the genome structure or different selective advantages of some segments may explain this complex recombination pattern.

Long-read sequencing further offered a unique opportunity to investigate recombination at the level of single viral genomes. Indeed, because Nanopore sequencing does not involve PCR or any amplification step, each read originated from a distinct DNA molecule, and an individual sequencing read could capture the recombination history of one distinct viral genome. The hCMV genome is ~235 kb long, and we recovered 59 reads longer than 200 kb. After manually analyzing SNPs on mapped reads, we reconstructed the recombination history of 38 viral genomes (Fig. 3f, Supplementary Fig. 6). Only three (8%) did not incorporate the gene drive cassette, and seven (18%) appeared to represent the pure TB40/E strain (the donor GD-mCherry viral genome). The remaining 28 sequences (74%) were recombination products of the Towne-eGFP and GD-mCherry genomes, and importantly, all included the gene drive cassette. Multiple genomes appeared to be exclusively from Towne origin, except for the incorporation of the gene drive cassette. On the other hand, some other sequences had incorporated major portions of both genomes. We also noted that most US regions originated from the TB40/E strain. These results demonstrated that the gene drive sequence can be efficiently and specifically transferred from the TB40/E strain to the Towne strain, creating new gene drive viruses. Homologous recombination appeared to be asymmetrically centered around the gene drive cassette, and created in some cases profound rearrangement between the two viral genomes. Thus, these data showed that gene drive virus could recombine with wildtype virus and incorporate the gene drive sequence into new viruses.

**Spread of gene drive sequences into the wildtype population.** We next aimed to show that the gene drive sequence could spread into a wildtype virus population. Fibroblasts were coinfected with wildtype Towne-eGFP (MOI = 0.1) and decreasing amounts of GD-mCherry. Viral titers in the supernatant of coinfected cells were measured by plaque assay and allowed us to follow the evolution of the viral population over time (Fig. 4a). We observed that, independently of the starting proportion of GD-mCherry (50%, 10%, or 0.1%), the gene drive cassette efficiently invaded the wildtype population and incorporated into most of eGFP expressing viruses. Viruses expressing both eGFP and mCherry indeed represented new gene drive viruses that kept propagating

the modification into the wildtype population. As could be expected, the time necessary for the gene drive to successfully invade the wildtype population was inversely correlated with the starting proportion of GD-mCherry: It was almost complete after 3 days when coinfecting with equal amount of the two viruses, but took nearly 2 months when starting with 0.1% of gene drive virus.

DNA double-strand break are typically repaired by two main mechanisms, homologous recombination and non-homologous end joining (NHEJ). Gene drives rely on homologous recombination to specifically introduce artificial sequences into unmodified genomes. By contrast, NHEJ repair is often imperfect and usually introduces small insertions and deletions. At the end of the coinfection experiments, we observed that a small proportion of GFP-only viruses remained (Fig. 4a). PCR and Sanger sequencing showed that the target site of most of these unconverted viruses had been mutated (Supplementary Fig. 7a, b). Interestingly, in every biological replicate tested, 40–60% of edited sequences had an identical 3 bp deletion centered around the cleavage site, highlighting that NHEJ is often not random. These edited GFP-only viruses have a mutated target site and have probably become resistant to the drive. To test this assumption, we isolated 12 GFP-only viral clones from three independent coinfection experiments and verified that they had a mutated target site (Supplementary Fig. 7c). Using six viral clones that had the same 3pb deletion, we repeated the coinfection experiment with GD-mCherry (Fig. 4b). In this experiment, only very few eGFP-mCherry recombinant viruses appeared, and GD-mCherry was unable to drive into the population of mutated GFP viruses. This showed that the GFP-only population present at the end of the experiment in Fig. 4a represent viruses that have become resistant to the drive.

Incidentally, this result further confirmed that a specific gRNA is necessary for a successful drive. Similarly, we barely observed the appearance of eGFP-mCherry recombinant virus when performing similar coinfection experiments with a *Cas9*-deleted version of the gene drive virus (GD-ΔCas9) (Fig. 4c). Moreover, because the Towne strain replicates faster than TB40/E (Fig. 2d), GFP-only viruses rapidly became dominant in these coinfection experiments where the drive could not occur (Fig. 4b, c, Supplementary Fig. 8). This makes the efficient introduction of the gene drive cassette into the Towne genome even more remarkable, since it is originating from a strain with a replicative disadvantage. Altogether, these experiments further proved that both Cas9 and a specific gRNA are necessary for the drive of the recombination cassette into the wildtype population.

**Spread of a defective gene drive virus.** We showed that the gene drive sequence can easily and efficiently spread into the wildtype population, representing up to 95% of the final proportion. However, a gene drive virus that replicates at levels similar to wildtype virus would have little therapeutic value. We wanted next to

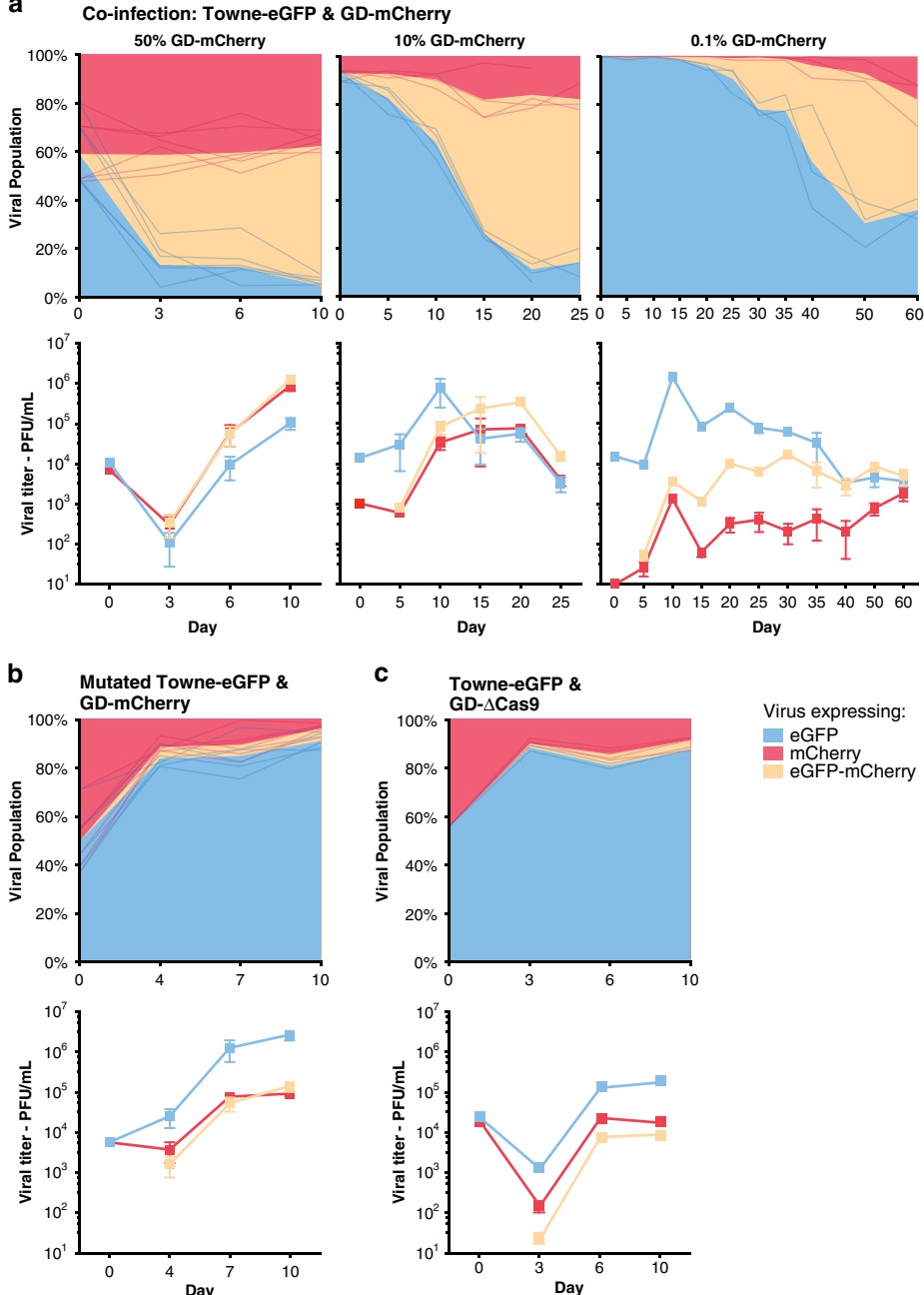

**Fig. 4 Gene drive sequences efficiently spread into the wildtype population. a** Coinfection experiments between Towne-eGFP (MOI = 0.1 at day 0) and different starting concentration of GD-mCherry: Viral titers (lower panels) and proportion (upper panels) over time of viruses expressing eGFP alone, mCherry alone, or both as measured by plaque assay. Left panel: 50% of GD-mCherry at Day 0, n = 6; middle: 10% at D0, n = 4; right: 0.1% at D0, n = 3. **b** Viral titers and proportion of viruses after coinfection with equal amount of Towne-eGFP (with a mutated target site) and GD-mCherry. n = 6. **c** Viral titers and proportion of viruses after coinfection with equal amount of Towne-eGFP and GD-ΔCas9. n = 3. Titers are expressed in plaque forming unit (PFU) per mL of supernatant. Error bars represent standard error of the mean (SEM) between biological replicates. In the panels representing the viral population, data show both the mean and the individual trajectory of biological replicates. Source data are provided as a Source Data file.

determine if a gene drive strategy could be used to limit or stop a viral infection. *UL23*, the viral gene targeted by our drive, was initially thought to be dispensable for hCMV replication in fibroblasts[16,23]. Serendipitously, it was later showed that UL23 is a tegument protein that blocks antiviral interferon-γ (IFN-γ) responses by interacting with human N-myc interactor (Nmi) protein[24]. As a result, the growth of *UL23*-knockout viruses is severely inhibited in infected cells treated with IFN-γ. In our system, the Cas9 target site is located immediately upstream of *UL23* coding sequence, and GD-mCherry viruses lack a *UL23* start codon

(Fig. 2b). As predicted, we observed that the replication of GD-mCherry virus was strongly and significantly inhibited when cells were cultivated in presence of increasing concentrations of IFN-γ, with a 250- and 8000-fold titer reduction with 10 and 100 ng/mL of IFN-γ, respectively (Fig. 5a, Supplementary Fig. 9a, b for thorough statistical analysis). Gene drive viruses that inactivate UL23 are, therefore, strongly attenuated by IFN-γ antiviral response, while wildtype viruses are minimally affected.

The unique sensitivity of GD-mCherry viruses to different concentrations of IFN-γ allowed us to test whether a severely

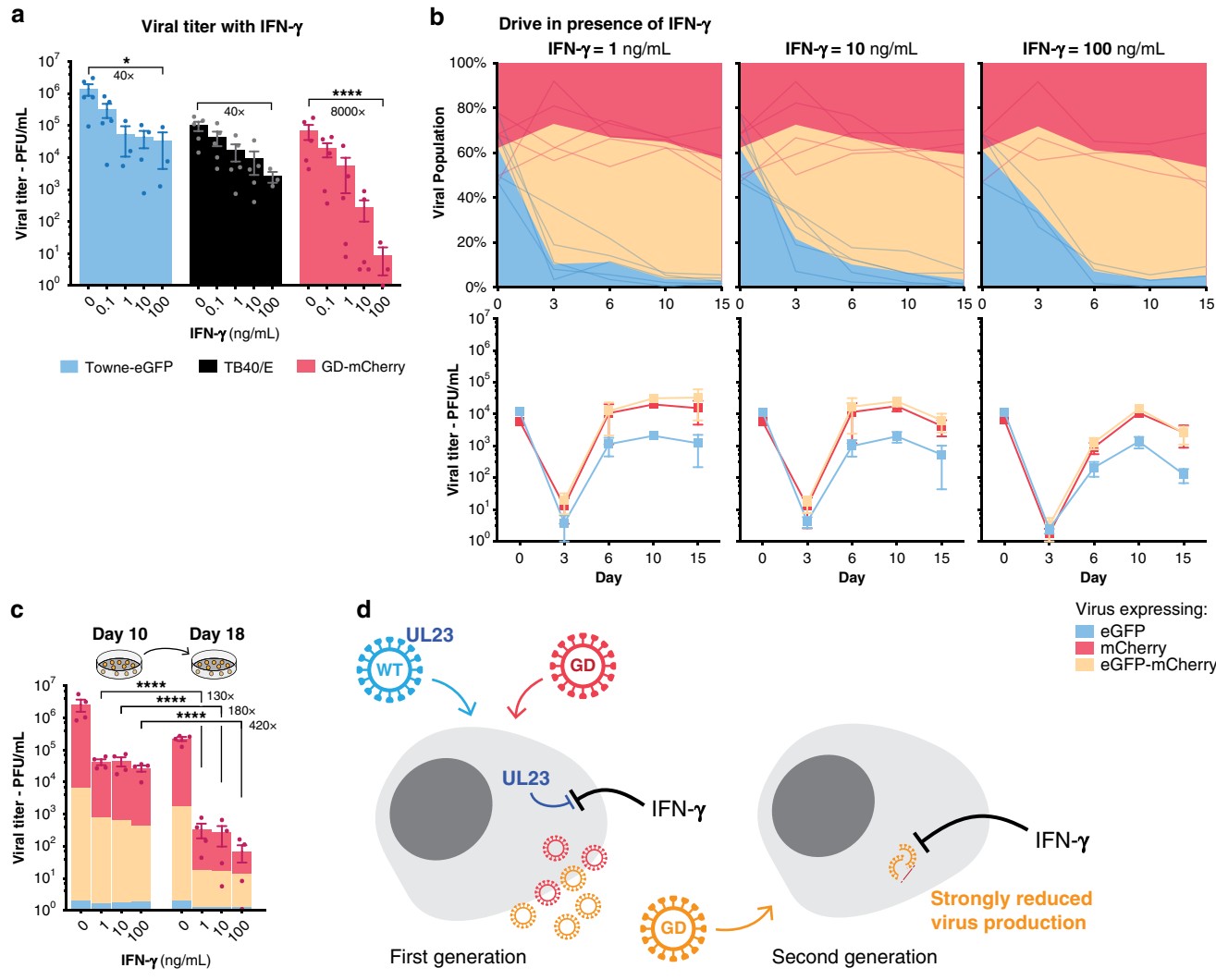

**Fig. 5 Spread of a defective gene drive virus. a** Viral titers at day 10 in the presence of increasing concentrations of Interferon-γ (IFN-γ). n = 3 (IFN-γ = 100 ng/mL) or n = 5 (other concentrations). **b** Viral titer and proportion of viruses expressing eGFP alone, mCherry alone, or both, after coinfection with equal amounts of Towne-eGFP and GD-mCherry, in presence of increasing concentrations of IFN-γ. n = 3 (IFN-γ = 100 ng/mL) or n = 5 (other concentrations). **c** Supernatants from coinfected cells were collected at day 10 and used to infect fresh cells. Titers were measured 8 days later. Colors indicate the proportion of the different viruses relative to the height of the bar. n = 4. **d** Model for the spread of defective gene drive viruses: Wildtype viruses express UL23 and block IFN-γ antiviral response while gene drive viruses are *UL23*-KO and are severely inhibited by IFN-γ. Upon coinfection, UL23 originating from the wildtype virus—either brought in with the incoming virion as a tegument protein or expressed early on—is sufficient to block the IFN-γ antiviral response. A first generation of new gene drive viruses can be created. These viruses are, however, *UL23*-KO and are severely inhibited by IFN-γ when infecting new cells. Titers are expressed in PFU/mL. Error bars represent SEM between biological replicates. *p-value < 0.05; ****p < 0.0001; two-way ANOVA with Sidak's multiple comparison test on log-transformed value. Source data are provided as a Source Data file.

defective virus could still efficiently drive into a wildtype population. We therefore repeated the coinfection experiment in the presence of increasing concentrations of IFN-γ (MOI = 0.1). We observed that the efficiency of the drive was unaffected by IFN-γ and that most Towne-eGFP viruses efficiently incorporated the gene drive cassette (Fig. 5b). After 10 or 15 days of coinfection, most of the viral population had been converted and would have been expected to show high IFN-γ susceptibility. Unexpectedly, however, viral titers reached levels similar to wildtype viruses after 10 or 15 days, even at IFN-γ concentrations where GD-mCherry viruses are normally totally defective (Fig. 5c). This suggested that wildtype viruses were able to complement defective ones, blocking IFN-γ response and allowing the spread of new gene drive viruses in the context of a coinfection system (model in Fig. 5d).

However, even if the first generation of gene drive viruses appeared to be rescued by wildtype viruses, we hypothesized that subsequent generations would be severely inhibited by IFN-γ. To test this hypothesis, supernatants of coinfected cells at day 10 were used to infect fresh cells, treated with similar concentrations of IFN-γ. Viral titers of this second generation of viruses were severely affected by IFN-γ, with titers dropping 180- or 420-fold with 10 and 100 ng/mL of IFN-γ, respectively (Fig. 5c). Our gene drive against *UL23*, therefore, allowed us to drastically suppress viral infection and, in fact, represents an ideal scenario. In the presence of IFN-γ, gene drive viruses are almost non-infectious and cannot propagate by themselves. However, coinfection with a wildtype virus efficiently complemented the defective viruses and allowed their replication at high titer. As a consequence, the modification was able to spread efficiently into the viral

population as long as wildtype viruses were present, but was unable to propagate further afterward (Fig. 5d). This represents an important example of how a viral gene drive could be used to limit a viral infection.

## Discussion

By using hCMV as a model, we offer here a proof of concept that a gene drive can be successfully designed for herpesviruses. A recent report described the design of a gene drive in bacteria, but this system was not intended to spread a trait into the bacterial population[25]. Our manuscript provides an example of a gene drive in a non-sexual organism that is able to spread a modification in the targeted population and replace their wildtype counterpart. We provide evidence that it is possible to design a drive that limits the infectivity of the virus. In the future, it will be interesting to determine if a similar viral gene drive can be designed against other herpesviruses or other DNA viruses and how the system can be adapted to function in an animal model and ultimately in human patients.

The technology relies on coinfection of cells by a wildtype and a modified virus, a condition that is easily satisfied in our cell culture model where the viral infection reaches a high MOI. Importantly, several lines of evidence indicate that coinfection also frequently happens in vivo in naturally occurring infections. First, recombination between different viruses is a well-known and widespread source of viral diversity in several herpesviruses including herpes simplex virus 1 (HSV-1)[26], Varicella-zoster virus[27] or hCMV[28]. Second, several studies that attempted to measure the rate of coinfection of herpesviruses in vivo could detect significant coinfection[29,30]. One study measured a level of 20–50% recombination between two HSV-1 viruses in a mouse model of viral encephalitis[29], and another detected a coinfection rate of 2–5% in different organs with mouse cytomegalovirus (mCMV)[30]. This second study with mCMV further showed that the replication of a defective virus could be complemented in trans by a wildtype virus, thereby validating our strategy in vivo (highlighted in Fig. 5d).

In order to gain better insight into what might happen under different coinfection rates, we performed numerical simulations to model gene drive spread into the viral population depending on two key variables: first, the fitness cost of the modified virus and second, the rate of coinfection. As expected, severely defective viruses did not spread into the wildtype population when the coinfection rate was very low (Supplementary Fig. 10b, upper left). However, with coinfection rates of 2–10%, a defective virus could spread in the population (Supplementary Fig. 10b, right panels). This level of coinfection corresponds to what has been observed experimentally in vivo[29,30] and suggest that a defective gene drive virus has the potential to spread in the viral population in vivo.

Herpesviruses establish life-long persistent infections in humans and animals. In immunocompetent hosts, they remain in a latent sate in the nucleus of infected cells indefinitely and only reactivate sporadically[1,31]. Given the low level of viral replication under these conditions, a gene drive would not be expected to spread in a predominantly latent infection. However, most severe diseases caused by herpesviruses are associated with persistent, non-latent infections, often in immunodeficient hosts. Important examples include viral encephalitis, infectious mononucleosis, Kaposi-sarcoma, or hCMV reactivation after organ transplant[32–34]. These represent attractive models where superinfection with gene drive viruses could potentially outcompete wildtype viruses and help clear out the infection. Herpesvirus are widespread in the human population and pre-existing immunity against herpesviruses could potentially prevent the successful superinfection with a gene drive version. However, there is abundant literature showing

that CMV or other herpesviruses can superinfect persistently infected immunocompetent hosts despite existing CMV-specific humoral and cellular immunity[28,35,36]. Besides, severe herpesvirus disease also predominantly affects immunocompromised patients. In this relevant context, it is possible that immune mechanisms would not be able to prevent superinfection and that a gene drive virus could help limit an infection. These considerations will have to be carefully examined in animal models.

The technology that we describe relies on homologous recombination. The high level of recombination that is naturally occurring in herpesviruses is probably one of the factors that made our system very efficient. However, recombination is normally a random process that exchanges fragments of herpesvirus genomes in an uncontrolled and unpredictable manner. By contrast, a gene drive specifically introduces a foreign sequence in a directed way. As we showed, the introduction and successful spread of a new trait in the population of viruses does not occur in the absence of Cas9 or a specific gRNA, and this is precisely the strength of the technique to render recombination precise, efficient and directed. Moreover, naturally occurring recombination only enables the selection and spread of variants that increase the fitness of the virus. By contrast, we showed that a gene drive can force the propagation of defective viruses, an observation which opens interesting therapeutic perspectives.

An important aspect of gene drive in sexually reproducing organisms such as mosquitos is the appearance and selection of drive-resistant alleles that can no longer be recognized by CRISPR gRNA[37,38]. Such alleles can already be present in the wild population, or appear when the target site is repaired and mutated by non-homologous end-joining instead of homologous recombination. These sequences are "immune" to future conversion into the drive allele, are often positively selected over time, and limit the ability to permanently modify a wildtype population[39–43]. Interestingly, we made very similar observations in our viral gene drive. We estimated that in our system ~90–95% of sequences were successfully converted by homologous recombination, but that the remaining viruses had become drive-resistant. It will be important to analyze in future experiments whether this drive-resistant population is positively selected over time and can outcompete gene drive viruses. This would not necessarily be a negative outcome, as it would ensure that gene drive viruses can be naturally contained. Moreover, our proof of concept targeted a viral gene, *UL23*, that is mostly dispensable in cell culture, and only shows a strong reduction in viral titer upon exogenous treatment with IFN-γ. Future gene drives should be designed against viral genes that cause a more constitutive defective phenotype, but still allow for an efficient spread. Ideally, the potential appearance and selection of drive-resistant viruses could be circumvented if the escapee mutations also rendered viruses non-replicative, for example if they knocked-out a conserved and critical viral gene, as it has been done in insects[44].

As a final note, the recent development of gene drives, notably in mosquitoes, have generated important ecological and biosafety concerns[45,46]. Our own work was conducted using laboratory viral strains unable to infect human hosts[47], and there were therefore no risks of inadvertent release of gene drive viruses into the wild. Our approach closely aligns with the guidelines established by the NIH and the National Academy of Science[48,49]. In the future, the risks and potential benefits of viral gene drives will need to be properly addressed and discussed with the scientific community.

## Methods

**Cells and viruses**. Human foreskin fibroblast cells were obtained from the ATTC (#SCRC-1041) and cultured in DMEM (10-013-CV, Corning, Corning, NY, USA), supplemented with 10% FBS (Sigma-Aldrich, St-Louis, MO, USA) and 100 μm/L

penicillin/streptomycin (Corning). Cells were regularly tested negative for myco-plasma and used between passages 3 and 13.

hCMV TB40/E-Bac4[17] and Towne-eGFP (T-BACwt)[18] were kindly provided by Edward Mocarski (Emory University, USA). To prepare viral stocks, cells were infected at low MOI (0.001–0.01) and kept in culture until 100% cytopathic effect was observed, usually after 10–15 days. Cells were then scraped out of the plate and centrifuged together with the supernatant (12,000×g, 1 h, 4 °C), resuspended in medium containing 5% milk, and sonicated to release cell-bound virions. Viral titers were assessed by plaque assay. Except when otherwise specified, subsequent infections were performed for 1 h at a MOI = 0.1, before replacing inoculum with fresh medium. Susceptibility to IFN-γ was assayed by virus growth in the presence of human recombinant IFN-γ (R&D, Minneapolis, MN, USA) after preincubation with IFN-γ for 2 h before infection.

Viral titers were assayed by plaque assay with 10-fold serial dilutions. 24-well plates were inoculated for 1 h and overlaid with 0.25% agarose. After 7–10 days, eGFP or mCherry fluorescent plaques were manually counted using an inverted microscope. Every viral plaque was analyzed on both green and red channel. 5–100 plaques were counted per well, and each data-point was the average of 3–4 technical replicates (i.e., 3–4 different wells).

Images of fluorescent viral plaques were acquired with a Nikon Eclipse Ti2 inverted microscope and Nikon acquisition software (NIS-Element AR 3.0). Channels for GFP and mCherry were merged and adjusted for contrast and exposure with ImageJ (v2.1.0).

Coinfection experiments were performed by coinfecting with wildtype Towne-eGFP and gene drive viruses for 1 h, with a total MOI of 0.1–0.2. For time-course experiments over multiple weeks, supernatants were used to inoculate fresh cells for 1 h before changing media.

### Cloning and generation of gene drive viruses.
A donor plasmid containing the gene drive cassette against *UL23* (GD-mCherry) between homology arms was generated by serial modifications of pX330, a codon-optimized *SpCas9* (from Streptococus pyogenes) and chimeric gRNA expression plasmid developed by the Zhang lab[50] (Addgene #42230). All modifications were carried out by Gibson cloning (NEB, Ipswich, MA, USA), using PCR products from other plasmids or synthesized DNA fragments (GeneArt™ String™ fragments, ThermoFisher, USA). Briefly, a fragment with a SV40 polyA terminator, a SV40 promoter and an mCherry fluorescent reporter was inserted between *SpCas9* and betaGlobin polyA signal. The PciI-AgeI fragment upstream of *SpCas9* was removed and replaced by *UL23* left homology arm (amplified from TB40/E-bac4). A fragment with UL23-5′ gRNA under a U6 promoter and UL23 right homology arm was finally inserted downstream of betaGlobin polyA signal between the NotI and XmaI restriction sites.

GD-ΔCas9 donor construct was subsequently generated by removing *SpCas9* by digestion and ligation. A donor construct to insert a SV40-driven mCherry reporter into the BAC cassette of hCMV TB40/E-bac4 was built similarly.

To build gene drive viruses, 1.5 million fibroblast cells were transfected with the homologous recombination donor plasmid and a helper plasmid (Addgene #64221)[51]. Transfection was performed by Nucleofection (Kit V4XP-2024, Lonza, Basel, Switzerland). 48 h after transfection, cells were infected for 1 h with hCMV TB40/E-bac4 at a low MOI (<0.1). After 7–10 days, viral plaques of mCherry-expressing cells were observed, suggesting successful integration of the gene drive sequence by homologous recombination (Supplementary Fig. 1a). mCherry-expressing viral plaques were isolated and purified by several rounds of serial dilutions and plaque purification. Purity and absence of unmodified TB40/E viruses were assayed by PCR after DNA extraction (DNeasy kit, Quiagen). PCR and Sanger sequencing across homology arms and cut sites confirmed that mCherry-expressing viruses contained the full gene drive sequence (Supplementary Fig. 1b). GeneRuler 1 kb DNA ladder (ThermoFisher) was used in agarose gels as a size marker. The ladder is shown in the gel images, with the stronger band corresponding to 500 bp.

Viral stocks were produced as specified above and titered by plaque assay.

Deconvolution of Sanger sequencing in Supplementary Fig. 7 was performed using Synthego ICE online tools (https://ice.synthego.com).

### HIRT DNA extraction and analysis of recombinant BAC clones.
hCMV epi-somal DNA was recovered from the whole population of coinfected cells by HIRT DNA extraction[19] 48 h after infection at a high MOI. Infected cells (grown in 1–2 T175 plates) were scraped-off, washed in PBS and resuspended in HIRT resus-pension buffer (10 mM Tris–HCl pH 8.0, 10 mM EDTA in 100 μL for 1 million cells). An equal volume of HIRT lysis buffer (10 mM Tris–HCl pH 8.0, 10 mM EDTA, 1.2% SDS) was added and gently mixed. NaCl was added (1 M final con-centration) before incubating overnight at 4 °C. Supernatant was collected after centrifugation (15,000×g, 20 min, 4 °C). Contaminating RNA was removed with RNAse A and hCMV DNA purified by double phenol–chloroform–isoamyl alcohol (25:24:1) extraction. DNA was precipitated with two volumes of pure ice-cold ethanol, washed with 70% ethanol, dried and resuspended in 10 mM Tris–HCl pH 8.0.

2–3 μL of recovered DNA was electroporated into NEB® 10-beta electrocompetent cells (C320K, NEB) and plated on chloramphenicol LB plates. BAC DNA were purified using ZR BAC DNA Miniprep Kit (Zymo Research, Irvine, CA, USA).

Presence of *mCherry* or *GFP* on recombinant BAC clones was confirmed by PCR. Homology arms of mCherry-eGFP clones were analyzed by PCR and Sanger sequencing using tools available on Benchling.com. Primers are given in Supplementary Table 1.

### Virion purification and Oxford nanopore sequencing.
Twelve large flasks were infected with recombinant viruses and cultured until the monolayer reached 100% cytopathic effect. Cells debris was pelleted away by centrifugation (20 min, 500×g, 4 °C) and supernatants were recovered. Virions present in the supernatant were pelleted by ultracentrifugation (22 krpm, 90 min, 4 °C, Beckman-Coulter rotor SW28) on a 5-mL cushion of 30% sucrose. Supernatants were discarded, and the pellets containing virions were resuspended in PBS (1 h at room temperature) and pooled in a final volume of 500 μL.

DNase I (20U, 30 min, 37 °C, NEB M0303S) and RNAse A (100 μg, 15 min, ThermoFisher EN0531) treatment removed contaminating human nucleic acids unprotected by the virus envelope and capsid. Viral envelopes were lysed and DNase I was inactivated by adding 5× lysis buffer for 10 min (0.5 M Tris pH 8, 25 mM EDTA, 1% SDS, 1 M NaCl. Incubation for 1 h at 55 °C with 8U of proteinase K (NEB P8107S) finally disrupted viral capsids. To recover full-length hCMV genomes, subsequent pipetting and centrifugation steps were performed extremely carefully with wide-bore pipet tips. hCMV DNA was purified by double phenol–chloroform–isoamyl alcohol (25:24:1) extraction and centrifugation (4000×g, 3 min). DNA was precipitated for 1 h at −80 °C with 1/20 volume of 5 M NaCl and 1.5 volume of cold ethanol, centrifuged (6800×g, 30 min, 4 °C), washed with 70% ethanol, dried and resuspended in 10 mM Tris–HCl pH 8.0 for 24 h at 4 °C without pipetting. DNA was quantified by Nanodrop.

Libraries were prepared using SQK-LSK109 ligation sequencing kit from Oxford Nanopore Technology (Oxford, UK), without any fragmentation steps, using wide-bore pipet tips and careful pipetting steps to minimize DNA shearing. Libraries for the first biological replicate (two technical replicates) were prepared following the manufacturer instructions, using 2 μg of starting material. Lambda DNA control was added in the first technical replicate. In an attempt to maximize read length, the second biological replicate was prepared with 15 μg of DNA, omitting the first AMPure XP bead clean-up. Sequencing was performed on two FLO-MIN106-R9 Flow Cells on a MinION Mk1B device following manufacturer instructions.

### Oxford Nanopore sequencing analysis.
Nanopore FAST5 raw data was con-verted into FASTQ files using Albacore v2.3.3 basecalling, and runs statistics were obtained using Nanoplot v1.0.0 (https://github.com/wdecoster/NanoPlot)[52]. Adaptors were removed with Porechop v0.2.4 (https://github.com/rrwick/Porechop). Reads with quality $Q > 6$ were filtered using Nanofilt v2.5.0 (https://github.com/wdecoster/nanofilt)[52] and Lambda DNA reads excluded with Nano-Lyse v1.0.0 (https://github.com/wdecoster/nanolyse)[52]. Technical replicates were merged for the rest of the analysis. Read lengths were filtered with Nanofilt.

Reference sequences for Towne-eGFP (GenBank KF493877) and TB40/E-Bac4 (GenBank EF999921) were downloaded. We first inserted the gene drive cassette into Towne-eGFP and TB40/E reference sequences. Reads were then mapped on a composite human hg38-Towne genome using Minimap2 v2.14 (https://github.com/lh3/minimap2)[53] and mapping statistics (Supplementary Fig. 4) were obtained with samtools v1.10[54] after filtering of secondary and supplementary reads (samtools −F 2048 −F 256).

To create of map of SNPs between Towne and TB40 strains, we created a FASTA file composed of multiple copies of the two genomes. Using Minimap2, this fasta file was mapped onto Towne-eGFP reference. SNPs were then called using bcftools mpileup and bcftools call, generating a BCF file with the SNPs coordinates:

```
minimap2 -a Ref_Towne.fasta multiple_Towne-TB40.fasta
-cs -N 0 -2 | samtools view -S -b| samtools sort -o mapped.bam
    bcftools mpileup -Ou -f Ref_Towne.fasta mapped.bam |
bcftools call -mv -Ob -o map_snp.bcf
```

hCMV genome exists in four different configurations depending on the respective orientation of UL and US segments. A Towne-eGFP composite genome composed of Towne sequences in the four configurations was finally created, and a complete map of SNPs in the four configurations was also generated.

After comparing different mappers, we mapped sequencing reads (length > 10 kb) on the composite Towne genome using Graphmap v0.3.0 (https://github.com/isovic/graphmap)[55], keeping only the best mapped reads (default, returning uniquely mapped read) or allowing multiple read mapping (option -Z). Proportion of variants for each SNP coordinate was then calculated using Nanopolish (https://github.com/jts/nanopolish)[56]. Nanopolish was run successively on the four subgenomes as follows, using the SNP map generated above (example for Towne in the SS: Sense–Sense configuration):

```
nanopolish variants−reads data.fastq.gz-bam
Mapped_with_graphmap.bam−genome Reference_Towne_4config.
fasta −p 2 −w Towne_config_SS:1-239862 -m 0.15 -x 2000 -c
map_snp_4config.vcf -o Call_SS.vcf
```

Read coverage and Support fractions were then extracted from the VCF files and plots were generated using R. Of note, for multiple mapping reads, we had to artificially increase the mapQ fields of SAM/BAM files by 20, because Nanopolish automatically discard such reads.

Finally, to reconstruct the recombination history of individual genomes, the longest reads (>200 kb) were mapped on the composite Towne genomes using Graphmap and visualized with IGV[57]. The recombination map of each individual read was then reconstructed manually using the SNP map.

The reference genome of GD-mCherry virus was inferred from reads that contained no Towne fragments. Importantly, we detected two large deletions in GD-mCherry not present in the original TB40E-bac4 genome: One 5.6-kb deletion in UL ranging from *RL12* to *UL8* genes and, therefore, including the frequently mutated *RL13* gene, and a second 4.6-kb deletion in US from *US15* to *US19*.

**Statistics and reproducibility**. Plaque assay data did not appear to satisfy the normality condition required for parametric tests. Due to small sample sizes, normality and lognormality tests could however not be performed. We therefore chose to run in parallel both parametric tests on log-transformed data, and non-parametric test on untransformed data, and reported results when both type of tests gave significant values. For groups analysis, we performed two-way ANOVA with Sidak's multiple comparison test on log-transformed data, and Kruskal–Wallis test with Dunn's multiple comparison test on untransformed data. Analysis were run using GraphPad Prism version 8.1.1 for macOS (GraphPad Software, San Diego, CA, USA, www.graphpad.com).

Examples of plaques shown in Fig. 3a and Supplementary Fig. 2 are representative of every plaque assay ($n > 100$) performed in the study.

**Numerical simulations**. Numerical simulations of viral gene drive were computed using a simplified viral replication model. Shortly, in each viral generation, N virtual cells were randomly infected and coinfected with N*MOI viruses, producing a new generation of viruses. In this new generation, wildtype viruses coinfected with drive viruses were converted to new gene-drive viruses. Gene drive virus replicate with a fitness cost *f*, and the coinfection rate is calculated from the MOI assuming a Poisson distribution. The code and a more thorough description are available at https://github.com/mariuswalter/ViralDrive.

**Reporting summary**. Further information on research design is available in the Nature Research Reporting Summary linked to this article.

## Data availability

The data supporting the findings of this study are available within the paper and its Supplementary Information files. Source files for Sanger sequencing or microscopy images are available upon request. Oxford nanopore Sequencing data have been deposited in the Short Read Archive with BioProject accession no. PRJNA545115. Plasmids, viruses and other reagents developed in this study are available upon request and subject to standard material transfer agreements with the Buck Institute. Source data are provided with this paper. Any other relevant data are available upon reasonable request.

## Code availability

Code developed for numerical simulations is available on GitHub (https://github.com/mariuswalter/ViralDrive).

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

## Acknowledgements

We thank Edward Mocarski (Emory University) for providing viral stocks. We thank members of the Verdin lab for technical and conceptual help. M.W. thanks Krystal Fontaine and Rosalba Perrone for their friendship and fruitful discussions. This study was funded through institutional support from the Buck Institute for Research on Aging.

## Author contributions

M.W. initiated and designed the study, performed all experiments and analyses. E.V. designed, supervised, and funded the project. M.W. and E.V. wrote the manuscript.

## Competing interests

A patent application describing the use of a gene drive in DNA viruses has been filed by the Buck Institute for Research on Aging (Application number PCT/US2019/034205, pending, inventor: M.W.). E.V. declares no competing interests.
