## [Peer Review File · Nature Communications]

Reviewers' Comments:

Reviewer #1:

Remarks to the Author:

The authors have made substantive changes and experimental additions to their manuscript. The story is both fascinating and better controlled than the original submission and clearly shows that CRISPR/Cas9-driven recombination enhances recombination rates sufficiently to be experimentally measurable as "gene drive".

Reviewer #4:

Remarks to the Author:

The authors were asked to demonstrate that gene drive expansion required a functional and specific gRNA. The suggested experiment was to test a gene drive virus with a non-specific or absent gRNA. Instead, the authors used a target virus with a 3bp deletion in the gRNA cutting site, and found it was resistant to the gene drive, and they use these data as evidence for the requirement of a specific gRNA. At first glance, the performed experiment seemed appropriate. However, in reading the paper it appears that the 3bp deletion virus was not de novo generated by the authors, but was selected as a virus resistant to the gene drive. So, what they have shown is that they can select a virus that is resistant to the gene drive, that it has a gRNA target site deletion, and that in subsequent experiments it remains resistant to the gene drive. But they have not shown that the target site deletion is the cause of resistance to the gene drive (I suspect it is, but that conclusion can't be made from the data presented). The authors either need to revert the 3bp deletion within the selected virus, or de novo generate the 3bp deletion target virus in a gene drive sensitive virus (or something similar, or one of the originally suggested viruses) and then test for gene drive sensitivity/resistance. The data they currently provide do not effectively address the major, unanimous criticism of the previously submitted work.

Minor comments:

Line 57: In the field, TB40/E is called a "clinical" strain and TOWNE is called a "laboratory" strain. If you want to avoid confusion, just say "infected with the TB40/E-bac4 strain of HCMV".

Line 90: "Data not shown" isn't allowed anymore, is it?

Reviewer #5:
Remarks to the Author:

To The Authors,

Summary:

Walter & Verdin describe a proof-of-concept gene-drive system in a DNA virus. In the manuscript, the authors build a gene-drive element (GD) inserted into the *UL23* locus and tagged with a Red fluorescent marker. They test the GD's ability to propagate into another viral line (Towne-eGFP) using the GFP marker to label the receiver strain. They performed co-infection of fibroblasts with both constructs and analyze HDR-based genotype conversion of the receiver virus by analyzing the expressed mCherry and GFP markers in re-infection experiments. Subsequently, the authors perform co-infections at different "wild-type" to gene-drive ratios and observe the capability of the built gene drive to spread within the wild-type population converting it to gene drive. The gene drive conversion events are confirmed by single-DNA-molecule sequencing of viral genomes and analyzing several isolates, allowing them to reconstruct several recombinant genomes. Importantly, the authors perform controls testing a Cas9 Δ gene-drive, which is unable to spread, as well as challenging a viral line containing a -3bp deletion at the target site with the gene-drive. In these experiments, they observe no-spread of the gene drive, and minimal conversion probably due to normal recombination during viral replication. Remarkably, in these experiments, the wild-type virus seems to be more "fit" than the gene drive strain, strengthening the authors' results regarding the gene drive spread. Lastly, they developed an elegant experiment highlighting that this gene drive could be developed in principle towards a therapeutic application.

Critique:

I believe this is a remarkable work that could be of interest to both the gene drive and virology fields, as well for the broader scientific community because of the novelty of applying a GD to a very different system, a virus. While this is a proof-of-principle work, it is not hard to imagine that some future iteration of this system could be developed into therapy. Since the gene drive field is also at its infancy (the majority of the work has been restricted to insects), this first application into a virus, while certainly warranting further investigation, represents a milestone in the field.

Comments:

I come from the gene-drive field and do not possess a strong virology background. Therefore I do not believe I could thoroughly review the methodologies and the specific details regarding the biology of the virus used.

It is my understanding that four other referees have reviewed this manuscript, and I agree with several of the comments that were raised before, such as the previous lack the control affecting gRNA function (-3bp strain challenged with GD). I believe that the authors have successfully addressed the previous issues raised by the referees and overall, I believe that the conclusions are now well-supported by the data presented in the manuscript.

I do have a few comments that I believe would further improve the manuscript:

1. Abstract/Lines 3-4: "*Current engineered gene drive strategies rely on sexual reproduction, and are thought to be restricted to sexual organisms*". I know the struggle of word limitation in abstracts. That said, it would be great to squeeze in a sentence explaining what a gene drive is and does, as a reader from outside the gene drive field, might not know the term.
2. Lines 89-90: "(data not shown)" I am guessing this refers to sequencing data pertaining to the experiments in Fig.3C? I believe it might be worth including these data as supporting information

as the recombination pattern of these recombinants might be very different from the one observed in Fig 2f.

3. Lines 175-178: It is very interesting how much the generation of resistant alleles parallels other fields. The prevalence of specific NHEJ has been described in the context of gene drive (Hammond et al. 2017 Plos Genetics, Lopez del Amo, et al. 2020 Nat Comms.) suggesting that similar mechanisms might be at play here. The authors could mention these previous results and discuss how theirs compare.
4. Lines 308-316: The authors could include some references here. My comment goes back to the fact that this manuscript bridges two very distinct fields and will have a broader readership that might be lacking a gene-drive or virology background (like me).

Response to reviewers

Manuscript NCOMMS-20-03617A

September 2020

Dear Referees,

We are thankful for the time and the thorough analysis of our work. We are very pleased that our work has now been accepted for publication. Please find a point by point response to the reviewer's comments below.

Sincerely,

The authors.

Point by point response to reviewers:

Reviewer #1 (Remarks to the Author):

The authors have made substantive changes and experimental additions to their manuscript. The story is both fascinating and better controlled than the original submission and clearly shows that CRISPR/Cas9-driven recombination enhances recombination rates sufficiently to be experimentally measurable as "gene drive".

- We are extremely grateful for this reviewer comment.

Reviewer #4 (Remarks to the Author):

The authors were asked to demonstrate that gene drive expansion required a functional and specific gRNA. The suggested experiment was to test a gene drive virus with a non-specific or absent gRNA. Instead, the authors used a target virus with a 3bp deletion in the gRNA cutting site, and found it was resistant to the gene drive, and they use these data as evidence for the requirement of a specific gRNA. At first glance, the performed experiment seemed appropriate. However, in reading the paper it appears that the 3bp deletion virus was not de novo generated by the authors, but was selected as a virus resistant to the gene drive. So, what they have shown is that they can select a virus that is resistant to the gene drive, that it has a gRNA target site deletion, and that in subsequent experiments it remains resistant to the gene drive. But they have not shown that the target site deletion is the cause of resistance to the gene drive (I suspect it is, but that conclusion can't be made from the data presented). The authors either need to revert the 3bp deletion within the selected virus, or de novo generate the 3bp deletion target virus in a gene drive sensitive virus (or something similar, or one of the originally suggested viruses) and then test for gene drive sensitivity/resistance. The data they currently provide do not effectively address the major, unanimous criticism of the previously submitted work.

- We agree that an experiment with a non-specific gRNA would have been the ideal control. However, because of the allotted time for the revision and the COVID pandemic, this was not experimentally feasible. We however believe that our approach using a 3pb deletion in the target site was appropriate, and addressed the initial reviewer's concerns.
- This reviewers write "But they have not shown that the target site deletion is the cause of resistance to the gene drive". We would like to point that 1). Target site mutation is the cause of resistance in all the documented gene drive designed in insects species. 2). To alleviate the concerns that the 3pb deletion could not be the cause of resistance, we performed the experiment in Fig. 4b with 6 resistant viruses that were independently isolated. If the 3pb deletion was not the cause of resistance, it would mean that in 6 independent viruses, other unknown mechanisms would have similarly evolved, which seems highly unlikely. 3). The point of the experiment was actually not to demonstrate that the site deletion was the cause of resistance, but that a non-specific gRNA would not mediate recombination, as we showed.

Minor comments:

Line 57: In the field, TB40/E is called a "clinical" strain and TOWNE is called a "laboratory" strain. If you want to avoid confusion, just say "infected with the TB40/E-bac4 strain of HCMV".

- This is a confusing comment as 'clinical' was removed from the revised manuscript.

Line 90: "Data not shown" isn't allowed anymore, is it?

- This was removed from the revised manuscript.

Reviewer #5

Summary:

Walter & Verdin describe a proof-of-concept gene-drive system in a DNA virus. In the manuscript, the authors build a gene-drive element (GD) inserted into the UL23 locus and tagged with a Red fluorescent marker. They test the GD's ability to propagate into another viral line (Towne-eGFP) using the GFP marker to label the receiver strain. The performed co-infection of fibroblasts with both constructs and analyze HDR-based genotype conversion of the receiver virus by analyzing the expressed mCherry and GFP markers in re-infection experiments. Subsequently, the authors perform co-infections at different "wild- type" to gene-drive ratios and observe the capability of the built gene drive to spread within the wild-type population converting it to gene drive. The gene drive conversion events are confirmed by single-DNA- molecule sequencing of viral genomes and analyzing several isolates, allowing them to reconstruct several recombinant genomes. Importantly, the authors perform controls testing a Cas9D gene-drive, which is unable to spread, as well as challenging a viral line containing a -3bp deletion at the target site with the gene-drive. In these experiments, they observe no-spread of the gene drive, and minimal conversion probably due

to normal recombination during viral replication. Remarkably, in these experiments, the wild-type virus seems to be more “fit” than the gene drive strain, strengthening the authors’ results regarding the gene drive spread. Lastly, they developed an elegant experiment highlighting that this gene drive could be developed in principle towards a therapeutic application.

Critique:

I believe this is a remarkable work that could be of interest to both the gene drive and virology fields, as well for the broader scientific community because of the novelty of applying a GD to a very different system, a virus. While this is a proof-of-principle work, it is not hard to imagine that some future iteration of this system could be developed into therapy. Since the gene drive field is also at its infancy (the majority of the work has been restricted to insects), this first application into a virus, while certainly warranting further investigation, represents a milestone in the field.

- We are extremely grateful for Valentino Gantz comments and we share his excitement!

Comments:

I come from the gene-drive field and do not possess a strong virology background. Therefore I do not believe I could thoroughly review the methodologies and the specific details regarding the biology of the virus used.

It is my understanding that four other referees have reviewed this manuscript, and I agree with several of the comments that were raised before, such as the previous lack the control affecting gRNA function (- 3bp strain challenged with GD). I believe that the authors have successfully addressed the previous issues raised by the referees and overall, I believe that the conclusions are now well-supported by the data presented in the manuscript.

I do have a few comments that I believe would further improve the manuscript:

1.

Abstract/Lines 3-4: “Current engineered gene drive strategies rely on sexual reproduction, and are thought to be restricted to sexual organisms”. I know the struggle of word limitation in abstracts. That said, it would be great to squeeze in a sentence explaining what a gene drive is and does, as a reader from outside the gene drive field, might not know the term.

- This is a good point and we modified the abstract and the text of the introduction to provide better context.

2.

Lines 89-90: “(data not shown)” I am guessing this refers to sequencing data pertaining to the experiments in Fig.3C? I believe it might be worth including these data as

supporting information as the recombination pattern of these recombinants might be very different from the one observed in Fig 2f.

- This was corrected in the revised manuscript

3.

Lines 175-178: It is very interesting how much the generation of resistant alleles parallels other fields. The prevalence of specific NHEJ has been described in the context of gene drive (Hammond et al. 2017 Plos Genetics, Lopez del Amo, et al. 2020 Nat Comms.) suggesting that similar mechanisms might be at play here. The authors could mention these previous results and discuss how theirs compare.

- We agree that the parallel is very striking! *Hammond et al. 2017* was already cited in our manuscript along with other sources (ref 37-43), and we devoted an entire paragraph in the discussion about this point. The evolution of resistance and the prevalence of NHEJ will be the subject of a follow-up study.
- *Lopez del Amo, et al. 2020* is a very interesting study (authored by this reviewer) that describes the design of a transcomplementing gene drive. We however feel this is not the most relevant study to refer to when discussing the apparition of resistance in gene drives.

Lines 308-316: The authors could include some references here. My comment goes back to the fact that this manuscript bridges two very distinct fields and will have a broader readership that might be lacking a gene-drive or virology background (like me).

- This is a good point and we included more references (ref 1,29-34).

Great work! Valentino Gantz.